# Ovarian Cancer in the Era of Precision Surgery and Targeted Therapies

**DOI:** 10.3390/cancers17203371

**Published:** 2025-10-18

**Authors:** Yagmur Sisman, Tim Svenstrup Poulsen, Tine Henrichsen Schnack, Claus Høgdall, Estrid Høgdall

**Affiliations:** 1Department of Gynecology, Copenhagen University Hospital, Rigshospitalet, 2100 Copenhagen, Denmark; yagmur.sisman@regionh.dk (Y.S.); claus.hogdall@regionh.dk (C.H.); 2Department of Pathology, Copenhagen University Hospital, Herlev University Hospital, 2730 Herlev, Denmark; tim.svenstrup.poulsen@regionh.dk; 3Department of Gynecology, Odense University Hospital, 5000 Odense, Denmark; tine.henrichsen.schnack@rsyd.dk

**Keywords:** ovarian cancer, precision surgery, high-grade serous carcinoma, targeted therapy

## Abstract

High-grade serous ovarian cancer is the most frequent and aggressive form of ovarian cancer. Although most patients respond to initial chemotherapy, relapses are almost inevitable. At relapse, surgery to remove recurrent disease can be considered in selected patients. In this study, we examined whether the genetic profile of the tumor changes from diagnosis to relapse. By analyzing paired tumor samples from 16 patients, we found that the mutational landscape remained highly stable, with only one case showing an additional mutation at relapse. These results suggest that the main genetic drivers are established early and persist throughout disease progression. Our findings emphasize the role of surgery in carefully chosen patients and indicate that mechanisms beyond genetic mutations should be explored to understand resistance and guide future treatment strategies.

## 1. Introduction

Managing relapsed high-grade serous ovarian cancer (HGSC) presents significant challenges. Historically, systemic platinum-based chemotherapy has been the cornerstone of treatment. Despite initial response, most of the patients with advanced-stage disease ultimately experience disease recurrence. Relapses can be diagnosed through radiographic progression, an increase in cancer antigen 125 (CA125), symptoms or clinical findings on physical examination.

The DESKTOP III trial demonstrated a significant survival benefit from cytoreductive surgery at relapse in selected patients with recurrence [1]. Patients with a positive Arbeitsgemeinschaft Gynäkologische Onkologie (AGO) score—defined by an Eastern Cooperative Oncology Group (ECOG) performance status 0, ascites < 500 mL, and complete resection at initial surgery were randomized to receive either cytoreductive surgery at relapse followed by chemotherapy or chemotherapy alone. The study demonstrated a significantly improved overall survival (OS) in the surgery arm (54 months vs. 46 months, HR 0.75, CI 0.59–0.96), with the most favorable outcome observed in patients who achieved complete resection (OS 62 months), while patients who underwent surgery with incomplete resection had a lower survival (OS 28 months) [1]. Two additional studies have also demonstrated a survival advantage in appropriately selected patients who achieved no residual disease following surgery. In the SOC1 study, cytoreductive surgery was associated with a significant improvement in progression-free survival (PFS), while in the GOG 213 trial, PFS improved from 16 to 22 months, although no OS benefit was observed [2,3]. All studies included patients with platinum-sensitive disease. Eligibility was based on a high likelihood of achieving complete cytoreduction—either assessed through validated predictive models or based on surgeon discretion. Current international guidelines now recommend that patients with a first relapse occurring more than six months after completion of platinum therapy should be referred to specialized gynecologic oncology centers for evaluation of potential cytoreductive surgery at relapse [4,5,6].

In parallel with advances in surgical strategies, recent years have seen a growing emphasis on targeted treatment approaches. Poly (ADP-ribose) polymerase (PARP) inhibitors have shown improved PFS and survival in recurrent platinum-sensitive, PARP inhibitor-naïve patients [7]. PARP inhibitor maintenance is currently recommended for PARP-naive patients with relapsed HGSC and a *BRCA1/2* mutation and/or HRD, following response to platinum-based chemotherapy [4,5,8]. Similarly, bevacizumab has shown benefit in prolonging PFS when combined with chemotherapy [9]. It should be offered to bevacizumab-naive patients with recurrent disease as part of a combined regimen followed by maintenance therapy [4,5,8].

To date, only a few biological drivers of recurrence and resistance have been identified, such as *BRCA1/2* reversion mutations or loss of *BRCA1* promoter methylation [10,11,12]. However, findings from the 2023 BriTROC-1 study, a UK-based translational effort, revealed striking genomic stability between diagnosis and relapse in HGSC. Among 276 women with relapsed HGSC, 134 had matched samples from both time points. Panel sequencing demonstrated high concordance, with only four discordant cases and no identified *BRCA1/2* reversion mutations [13].

In this study, we aimed to assess differences in molecular driver mutations and therapeutic targets in tissue samples from diagnosis and relapse in 16 patients with HGSC undergoing cytoreductive surgery, alongside an evaluation of treatment patterns and survival outcomes. Our findings should be interpreted with caution due to the small sample size and warrant confirmation in larger patient cohorts.

## 2. Materials and Methods

### 2.1. Patients and Tissue Samples

Initially, 23 patients with HGSC were enrolled. Four were excluded due to insufficient tumor material, and three were excluded following quality assessment for sequencing due to low tumor content, resulting in a final cohort of 16 patients who underwent cytoreductive surgery at relapse. The included and excluded patients differed in the type of surgery. Thus, 86% of excluded vs. 19% of included cases underwent interval debulking surgery. No other clinical differences were identified between the groups (Appendix A). DNA sequencing was performed on all patients using tissue samples obtained during cytoreductive surgery at both diagnosis and relapse. Clinical data were obtained from each patient’s medical record. Platinum sensitivity was defined as the absence of relapse or progressive disease within 12 months after completion of first-line platinum-based chemotherapy. Partial platinum sensitivity was defined as relapse occurring between 6 and 12 months after treatment completion. Relapse was defined based on the best clinical evaluation, including CT or PET-CT imaging, serum CA125 levels, and patient symptoms. Overall survival was not reported, as only two patients had died at the time of data cut-off. Instead, time to first relapse and total follow-up time were chosen as clinically meaningful endpoints. All patients were enrolled in the Pelvic Mass Study/GOVEC Study and followed from March 2014 until May 2025. No patients were lost to follow-up. All tissue samples were registered and stored in the Danish Cancer Biobank. The diagnosis of HGSC was confirmed by an experienced gynecologic oncology pathologist through a review of the original tissue samples.

### 2.2. DNA Extraction, Library Preparation and Sequencing

A total of 32 samples were collected. A representative area of cancerous tissue was identified and collected as fresh-frozen tissue in 17 samples. In 15 samples, a representative area within FFPE blocks was identified and extracted to ensure a high tumor cell content. All assessments were performed by an experienced gynecologic oncology pathologist. Genomic DNA was extracted using the Maxwell^®^ RSC DNA FFPE Kit (Promega, Madison, WI, USA). DNA concentration was measured using the Qubit™ dsDNA High Sensitivity Assay Kit (Thermo Fisher Scientific, Waltham, MA, USA) on the Qubit Fluorometer (Thermo Fisher Scientific, Waltham, MA, USA). Library preparation for the OCA- Plus panel, covering 501 cancer-related genes, was performed manually. Sequencing was performed on the Ion S5™ XL Sequencer (Thermo Fisher Scientific, Waltham, MA, USA), and data were analyzed using Ion Reporter™ Software (v. 5.14) and the Oncomine Comprehensive Plus–w2.0–DNA–Single Sample workflow [14]. Only pathogenic and likely pathogenic variants, defined by ClinVar and classified according to AMCG criteria, were included for further analyses [15]. All performed molecular analyzes are somatic and not germline.

### 2.3. Data Analyses

All data analyses were carried out using Ion Reporter Software version 5.20 (Thermo Fisher Scientific, Waltham, MA, USA). The analyses were conducted using the human hg19 as the standard reference genome. The workflow Oncomine Comprehensive Plus-w2.3-DNA-Single Sample was utilized to identify somatic variants [single nucleotide variation (SNV), multi nucleotide variation (MNV), insertion-deletion (indel), and copy number variation (CNV)]. The Ion reporter’s coverage analysis reports were also used to evaluate the quality of the sequencing reactions by measuring mapping reads, mean depth, uniformity, and alignment over a target region. The genetic variants were clinically annotated using the database provided at VarSome using the implemented ClinVar database. Only variants classified as pathogenic and likely pathogenic (class 4 or 5) according to ACMG variant classification were included in the analysis [15].

### 2.4. Statistical Analyses

All statistical analyses were performed with Rstudio version 2025.05.1-513. Survival outcomes were summarized descriptively using medians and ranges. Formal subgroup comparisons (e.g., *BRCA*-mutated vs. wild-type) were not performed, as the study was not powered to detect statistically significant differences. Reported subgroup outcomes should therefore be considered exploratory and descriptive. No formal power calculation was performed prior to this study. The analyses were exploratory in nature, and the sample size was determined by the availability of patients meeting the inclusion criteria.

## 3. Results

### 3.1. Patients

Clinical characteristics of the 16 included HGSC patients at the time of cytoreductive surgery, both at diagnosis and relapse, are summarized in Table 1. All patients underwent cytoreductive surgery at diagnosis. At first relapse, 14 patients (87.5%) underwent cytoreductive surgery, and 2 patients (12.5%) underwent surgery at second relapse (Figure 1). The median age at diagnosis was 64 years, increasing to 67 years at follow-up. At the time of cytoreductive surgery at relapse, 14 patients (87.5%) had performance status 0, while the remaining 2 patients (12.5%) had a performance status of 1. Fourteen patients (87.5%) were platinum-sensitive, and two patients (12.5%) were partial platinum-sensitive. Complete resection was achieved in all patients during surgery at diagnosis. At cytoreductive surgery for relapsed disease, 14 patients (87.5%) had complete resection, whereas 2 patients (12.5%) were unresectable. The median follow-up was 63 months. The median time from diagnosis to first relapse was 32 months. The median follow-up for patients with complete resection was 64 months, compared to 33 months for those deemed unresectable. At time of follow-up, 14 patients (87.5%) were alive (Figure 1).

### 3.2. Treatments

At diagnosis, primary cytoreductive surgery was performed in 13 patients (81%), and interval debulking surgery in 3 patients (19%). Four patients (25%) had a third cytoreductive surgery, and one patient underwent a fourth cytoreductive surgery at the time of follow up (Figure 1).

All patients received platinum-based chemotherapy at both initial diagnosis and after surgical treatment of their first or second relapse. Chemotherapy regimens administered at subsequent relapses are detailed in Appendix A.

Seven patients (44%) received bevacizumab as part of their treatment plan. The timing and duration of the individual bevacizumab treatments are presented in Appendix A.

PARP inhibitors were part of the treatment plan for ten patients (63%). The timing and duration of the individual PARP inhibitor treatments is presented in Appendix A.

### 3.3. Pathogenic and Likely Pathogenic Mutations

All patients harbored pathogenic or likely pathogenic mutations (Table 2). *TP53* mutations were identified in 14 patients (88%), and *BRCA1/2* mutations were present in 6 patients (38%). Only one patient (6%) showed a difference in the mutational profile between the samples obtained at diagnosis and cytoreductive surgery at relapse. In this case, a newly acquired mutation in the *NOTCH2* gene was detected in tissue from relapse (Table 2).

At cytoreductive surgery at diagnosis, carcinomatosis was present in all *BRCA1/2* mutated patients and in 6 (60%) of *BRCA1/2* wild-type patients (Appendix A). Conversely, at cytoreductive surgery at relapse, carcinomatosis was observed in only 1 (17%) of BRCA1/2-mutated patients compared to 6 (60%) of BRCA1/2 wild-type cases. Of the four patients without carcinomatosis at diagnosis (patient ID 9, 10, 13, and 19), only one (patient ID 19) remained free of carcinomatosis at the time of cytoreductive surgery for relapse. Apart from the observation that the majority of *BRCA1/2* mutated patients (83%) did not present with carcinomatosis at relapse, no other associations between mutations and the presence of carcinomatosis could be identified. Complete resection at cytoreductive surgery for relapse was achieved in all *BRCA1/2* mutated patients, whereas two (20%) of the *BRCA1/2* wild-type patients were unresectable. Median time to first relapse was prolonged in the *BRCA1/2* mutated group (41 months; range 9–66) as compared with the *BRCA1/2* wild-type group (26 months; range 14–55). Overall follow-up was likewise longer for *BRCA1/2* mutated patients (median 96 months; range 63–121) than for *BRCA1/2* wild-type patients (median 46 months; range 33–75).

## 4. Discussion

In this study with 16 patients with HGSC who underwent cytoreductive surgery at diagnosis and relapse, 32 paired tissue samples were analyzed. The mutational landscape remained largely stable over time. All patients harbored pathogenic or likely pathogenic mutations at diagnosis, and in 15 of 16 cases (94%) the mutational profile was unchanged at relapse. Only one patient acquired an additional mutation, indicating that mutational alterations remained remarkably consistent despite disease progression and treatment.

The drivers of recurrence and resistance in HGSC remain poorly understood, and no biomarker has yet entered clinical practice. We collected tumor biopsies at diagnosis and relapse and observed no significant changes in mutations over time. Our findings contribute to the discussion about whether routine biopsies for molecular analyses in the relapse setting should be implemented in clinical practice, as no new actionable mutations emerge over time [13].

Similarly, the BriTROC-1 study, which included tumor biopsies from 276 patients with relapsed HGSC, demonstrated high concordance between primary and relapse samples, with only four discordant cases [13]. These included mutations unique to either diagnosis or relapse, but no consistent acquisition of new driver events. The study also reported stability in copy number alterations, suggesting that genetic evolution is not the main driver of relapse. While BriTROC-1 included both platinum-resistant and platinum-sensitive patients and used both surgical and image-guided biopsies, our study focused exclusively on a surgically treated, platinum-sensitive cohort, offering a clinically well-defined comparison.

The OCTIPS consortium analyzed 31 paired samples of platinum-sensitive HGSC using whole-exome sequencing and SNP array profiling [16]. Although differences were observed between primary and relapse tumors, no recurrent genomic alterations were consistently identified across pairs. Taken together, our findings, along with those from BriTROC-1 and OCTIPS, indicate that acquired resistance in HGSC is not driven by new mutational events. Instead, other mechanisms—such as epigenetic modulation, transcriptional reprogramming, or microenvironmental factors—are likely to underlie treatment resistance [13,16]. Notably, previous research has demonstrated that loss of *BRCA1* promoter methylation can contribute to resistance to both platinum-based chemotherapy and PARP inhibitors [11,12].

Systemic therapy has traditionally been the standard for recurrent HGSC, but the DESKTOP III trial demonstrated that cytoreductive surgery at relapse in addition to platinum-based chemotherapy improves OS, particularly when complete resection is achieved [1]. Although OS could not be estimated in our cohort as most patients were still alive, the median follow-up of 63 months and a complete resection rate of 88% support the survival benefit observed in DESKTOP III. Moreover, they found that patients who achieved complete resection had the most favorable outcomes. These findings are consistent with ours, where the median follow-up time for patients with complete resection was 64 months, compared to 33 months for those deemed unresectable [1].

The novelty of our study lies in the specific clinical context: we investigated a surgically treated, platinum-sensitive cohort with high rates of complete resection at relapse and prolonged follow-up. This design allowed us to analyze paired high-quality tumor samples from both diagnosis and relapse, within a population characterized by extended disease control and favorable overall survival. By showing that mutational stability persists, our study highlights that mechanisms of recurrence and resistance are unlikely to be explained by new somatic mutations. These findings have important clinical implications. The observed mutational stability suggests that repeated mutational profiling at relapse may offer limited additional value in this setting. Instead, research and clinical efforts could be more effectively directed toward understanding alternative mechanisms of treatment resistance, including epigenetic modulation, transcriptional reprogramming, and changes in the tumor microenvironment.

We observed a trend toward less carcinomatosis at relapse among *BRCA* mutated patients (17% vs. 60%), although this finding is based on very small numbers and should be regarded as hypothesis-generating only. Previous studies have not demonstrated such an association, and in our cohort all *BRCA* mutated patients were completely resected, compared with 80% in the BRCA wild-type group. One study, which aimed to examine whether *BRCA* mutated patients were more likely candidates for cytoreductive surgery at relapse, found no difference in the number of lesions at recurrence between *BRCA* mutated and *BRCA* wild-type patients [17]. Furthermore, they did not find a higher rate of complete resection among *BRCA* mutated patients, whereas in our study, all *BRCA* mutated patients achieved complete resection compared to 80% in the *BRCA* wild-type group. Notably, the two *BRCA* wild-type patients who were unresectable both presented with carcinomatosis. Carcinomatosis has previously been identified as a negative predictive of complete resection [18,19,20]. In the study by Harter et al., only 26% of patients with carcinomatosis achieved complete resection compared with 74% without, although survival did not differ if resection was complete [18]. The role of tumor biology remains to be clarified in the recurrent setting. It has long been unclear whether improved resectability—and consequently, better survival—reflects a more favorable tumor biology, or whether poor surgical outcomes are merely a consequence of a more aggressive disease less amenable to complete resection [21,22]. Our findings raise the question of whether BRCA status may indirectly influence surgical outcomes through underlying tumor biology. Future research integrating genomic, radiologic, and clinical variables could help elucidate whether BRCA status contributes to predictive models for resectability or treatment response in recurrent HGSC.

Our study has several methodological and clinical strengths. By combining longitudinal clinical data with sequencing of paired tumor samples from diagnosis and relapse, we could track mutational changes over and confirm that key driver mutations remain stable. The cohort represents a clinically relevant group —patients selected for cytoreductive relapse surgery—in whom molecular data are sparse. Rigorous pathological review ensured high tumor content, and variant interpretation followed ACMG and ClinVar criteria, supporting reproducibility [15]. With a median of 63 months, a high complete resection rate (87.5%), and integration in national biobanking programs, the study provides robust and well-annotated material for translational research.

This study has several limitations. The small and highly selected cohort limits statistical power and generalizability. The limited sample size also constrains the ability to detect differences and to draw firm conclusions from subgroup analyses, such as those comparing BRCA-mutated and BRCA wild-type patients. Moreover, as all patients in this study underwent cytoreductive surgery at relapse, the findings reflect a surgically favorable and platinum-sensitive population and should not be extrapolated to patients with unresectable or platinum-resistant disease. Therefore, these findings should be interpreted as hypothesis-generating and validated in larger, more diverse cohorts. Furthermore, the cohort represents a highly selected group of patients who underwent cytoreductive relapse surgery, which does not reflect the broader HGSC population, particularly those with unresectable disease. Accordingly, our conclusions should be viewed with caution and primarily as exploratory observations that warrant confirmation in prospective studies including less selected patient groups. As all patients were treated at a single national institute, local clinical practice may also influence generalizability, although standardized procedures can be considered a strength. Moreover, treatment algorithms and criteria for secondary cytoreductive surgery may differ across countries, which could affect patient selection and outcomes. However, as Denmark follows international ESMO–ESGO guidelines for recurrent ovarian cancer, we consider our findings broadly comparable to other Western healthcare settings.

Many excluded patients had undergone interval debulking surgery, where neoadjuvant chemotherapy often reduces tumor content, making molecular analyses difficult [23,24]. This challenge has also been reported in other studies, and while we observed a higher proportion of interval surgery among excluded patients, no other systematic clinical differences were seen [23]. Previous studies have demonstrated that neoadjuvant chemotherapy followed by interval debulking surgery and completion of chemotherapy is not inferior to primary cytoreductive surgery followed by chemotherapy in patients with FIGO stage IIIC or IV disease, where complete resection at primary surgery is unlikely or where the patient is unfit for extensive surgery [25,26]. Given these findings, it is unlikely that our exclusion of a higher proportion of interval debulking surgery patients introduced a survival-related selection bias [27]. However, we acknowledge that excluding cases with insufficient tumor tissue may have introduced a potential selection bias, as these patients could differ biologically from those with adequate tissue for analysis.

Additionally, we lack data on patients who were offered but not included for relapse surgery, which prevents us from fully assessing selection bias [27]. Thus, our findings may not be representative of all patients with relapsed HGSC, especially if those included had disease that was more accessible for resection.

Finally, we applied a 501-gene cancer panel, as it covers the clinically relevant and currently targetable variants, and allowed us to obtain higher sequencing depth in paired tumor samples [28,29]. Nevertheless, we acknowledge that broader sequencing strategies, such as WES or WGS, may reveal additional alterations of potential relevance. However, since some of the tumor samples were FFPE, these approaches were not feasible. Future studies with larger patient cohorts would therefore benefit from applying more comprehensive approaches to further explore potential mechanisms of platinum resistance.

Our findings provide insights into a selected group of patients undergoing cytoreductive surgery for relapsed HGSC, but validation in larger, prospective cohorts is needed. Future studies should confirm the observed mutational stability, assess molecular predictors of surgical outcome, and test whether *BRCA* status influences disease spread at relapse. Such work should include both resectable and unresectable patients, apply standardized tissue-collection protocols, and incorporate multi-omic profiling to capture genomic, epigenetic, and transcriptomic mechanisms of resistance. Deep clinical annotation, long-term follow-up, and patient involvement in study design will be essential to maximize translational relevance [11,12,30]. In summary, our study underscores that HGSC recurrence occurs without major mutational evolution, highlighting the importance of integrating molecular stability into future treatment strategies and trial design.

## 5. Conclusions

The somatic mutational landscape remains largely unchanged in patients with relapsed HGSC undergoing cytoreductive surgery at relapse, supporting the notion that key driver mutations emerge early and persist throughout disease progression. Clinically, high rates of complete resection and long-term survival were demonstrated, reinforcing the benefit of cytoreductive surgery at relapse in well-selected patients. Future prospective, multi-institutional studies are warranted to validate these observations and further refine patient selection.

## Figures and Tables

**Figure 1 cancers-17-03371-f001:**
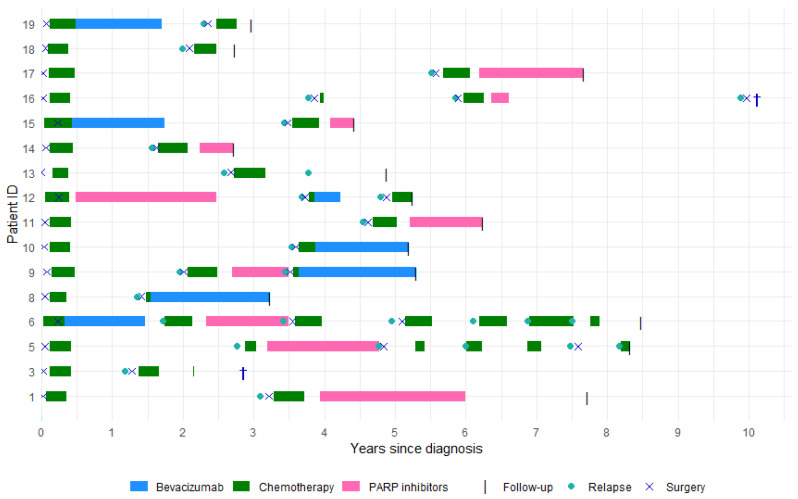
Overview of treatment timelines and clinical events in 16 HGSC patients.

**Table 1 cancers-17-03371-t001:** Clinical characteristics of 16 platinum-sensitive relapsed HGSC patients from a cohort of EOC patients undergoing relapse surgery at the University Hospital of Copenhagen.

	Diagnosis	Secondary Cytoreductive Surgery
Age in years median (range)	64 (41–74)	67 (46–77)
CA125 median (range)	195 (14–8890)	35 (6–490)
BMI median (range)	24 (18–34)	24 (18–35)
**P** **erformance score**		
0	9 (56%)	14 (87.5%)
1	6 (38%)	2 (12.5%)
2	1 (6%)	-
**FIGO stage**		
I-II	4 (25%)	-
III-IV	12 (75%)	-
**Surgery**		
Primary debulking	13 (81%)	-
Interval debulking	3 (19%)	-
**Residual tumor after surgery**		
0	16 (100%)	14 (87.5%)
Unresectable	0	2 (12.5%)
**Platinum response**		
>12 months (sensitive)	14 (87.5%)	-
6 – ≤ 12 months (partial sensitive)	2 (12.5%)	-
**Follow-up in months**		
Time to first relapse median (range)	-	32 (14–66)
Total follow-up time median (range)	-	63 (33–122)

BMI: Body mass index, CA125: Cancer Antigen 125, EOC: Epithelial ovarian cancer, FIGO: International Federation of Gynecology and Obstetrics, HGSC: High grade serous carcinoma.

**Table 2 cancers-17-03371-t002:** Pathogenic and likely pathogenic mutations at diagnosis and cytoreductive relapse surgery.

**Patient ID**	**Diagnosis**	**Relapse**
1	*B**RCA2* (c.7876T>C p.Trp2626Arg exon 17) *TP53* (c.422G>A p.Cys141Tyr exon 5)	*BRCA2* *TP53*
3	*TP53* (c.810T>G p.Phe270Leu exon 8)	*TP53*
5	*BRCA2*(c. 37G>T p.Glu13Ter exon 2)	*BRCA2*
6	*BRCA1* (c.115T>G p.Cys39Gly exon 3) *TP53* (c.1028delA p.Glu343GlyfsTer2 exon 10)	*BRCA1* *TP53*
8	*FANCD2* (c.935delT p.Leu312CysfsTer10 exon 12)	*FANCD2*
9	*ILR7* (c.636T>A p.Tyr212Ter exon 5) *TP53* (c.659A>G p.Tyr220Cys exon 6)	*ILR7* *TP53* ***NOTCH2*** (c.6575_6605del p.Pro2192LeufsTer20 exon 34)
10	*TP53* (c.916C>T p.Arg306Ter exon 8)	*TP53*
11	*TP53* (c.839G>A p.Arg280Lys exon 8)	*TP53*
12	*BRCA2* (c.5903C>G p.Ser1968Ter exon 11) *TP53* (c.150_151insT p.Glu51Ter exon 4)	*BRCA2* *TP53*
13	*TP53* (c.536A>G p.His179Arg exon 5)	*TP53*
14	*TP53* (c.742C>T p.Arg248Trp exon7)	*TP53*
15	*TP53* (c.734G>A p.Gly245Asp exon 7)	*TP53*
16	*BRCA2* (c.5346G>A p.Trp1782Ter exon 21) *RB1* (c.37_40delGCCG p.Ala13ProfsTer51 exon1) *TP53* (c.339_341delCTT p.Phe113del exon 4)	*BRCA2* *RB1* *TP53*
17	*BRCA2* (c.2454delT p.Gln819LysfsTer6 exon 11) *TP53* (c.731G>A p.Gly244Asp exon 7)	*BRCA2* *TP53*
18	*TP53* (c.659A>C p.Tyr220Ser exon 6)	*TP53*
19	*TP53* (c.1109C>T p.Arg337Cys exon 10)	*TP53*

*BRCA1:* Breast cancer gene 1, *BRCA2:* Breast cancer gene 2, *FANCD1:* Fanconi Anemia Group D2, *ILR7:* Interleukin-7 receptor, *RB1:* Retinoblastoma 1, *TP53:* Tumor Protein P53.

## Data Availability

Data are contained within the article.

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
