# Peer review of "Ovarian Cancer in the Era of Precision Surgery and Targeted Therapies"

_cancers, 2025, doi:10.3390/cancers17203371_

Round 1
Reviewer 1 Report
Comments and Suggestions for Authors
The manuscript presents a retrospective study on high-grade serous ovarian cancer, focusing on whether the mutational profile of tumors changes between diagnosis and relapse in patients undergoing cytoreductive surgery. Sixteen patients with paired tumor samples were sequenced using a 501-gene panel. The authors found remarkable genomic stability, with TP53 and BRCA1/2 mutations being the most common. Only one patient acquired an additional mutation (NOTCH2) at relapse. Clinically, the cohort demonstrated high rates of complete resection (88%) and favorable survival outcomes, with most patients alive at long-term follow-up. The study suggests that recurrence and treatment resistance in HGSC are unlikely to be driven by new mutational events and underscores the role of cytoreductive surgery in carefully selected patients.
Major Comments
Novelty and Significance: The study confirms prior large-scale observations (e.g., BriTROC-1) of mutational stability in HGSC but adds value by focusing on a surgically treated, platinum-sensitive cohort with long follow-up. However, the novelty should be framed more explicitly—what new insight does this cohort provide beyond confirming stability?
Methods:
Clarify the rationale for using a 501-gene panel and whether more comprehensive approaches (e.g., WES, WGS) might have detected other relevant alterations.
Provide additional details on sample selection and quality control, especially regarding excluded cases, to better understand potential biases.
Statistical analysis is described but limited—consider clarifying whether survival comparisons (BRCA-mutated vs. wild-type) were formally tested.
Results:
Results are clearly presented, but the small sample size (n=16) should be emphasized earlier as a key limitation.
The clinical outcomes section would benefit from more quantitative survival data (median OS, if estimable, or landmark survival rates).
The observation that BRCA-mutated patients had less carcinomatosis at relapse is interesting but underpowered; it should be framed strictly as hypothesis-generating.
Discussion:
The discussion is thorough but could be more concise, as some sections repeat background information.
Expand on the clinical implications: Should mutational profiling at relapse be deprioritized, and should resources shift toward studying epigenetic or non-genetic resistance mechanisms?
The potential role of BRCA status in surgical selection is intriguing but requires cautious wording; suggest adding more context on how this could inform future predictive models.
Limitations: the following should be stressed more:
Small, highly selected cohort—results may not generalize to unresectable patients.
Use of targeted sequencing rather than broader genomic or epigenomic profiling.
Potential bias from excluding cases with insufficient tumor tissue.
Minor Comments
Abstract: Could highlight the clinical takeaway more clearly (e.g., stability of mutations suggests limited utility of repeat sequencing at relapse).
This is a well-conducted and carefully reported study that adds meaningful data on mutational stability in HGSC and long-term outcomes after cytoreductive surgery at relapse. With revisions to clarify novelty, strengthen methodological transparency, and streamline the discussion, the manuscript will make a valuable contribution.
Comments on the Quality of English LanguageMinor editing required to make phrases more fluid
Author Response
Major Comments
Novelty and Significance: The study confirms prior large-scale observations (e.g., BriTROC-1) of mutational stability in HGSC but adds value by focusing on a surgically treated, platinum-sensitive cohort with long follow-up. However, the novelty should be framed more explicitly—what new insight does this cohort provide beyond confirming stability?
We appreciate this thoughtful comment. While our findings are in line with prior large-scale studies, we believe the novelty of our work lies in the focus on a surgically treated, platinum-sensitive cohort with long follow-up. This setting provides additional clinical context, showing that mutational stability also characterizes patients with prolonged disease control and favorable surgical outcomes. We have now clarified this point in the Discussion (lines 277-283).
Methods:
Clarify the rationale for using a 501-gene panel and whether more comprehensive approaches (e.g., WES, WGS) might have detected other relevant alterations.
We thank the reviewer for this valuable comment. We chose the 501-gene panel because it includes the clinically relevant and targetable variants known today, and our initial focus was to evaluate whether mutational changes occurred that could render patients eligible for targeted treatment options. In addition, this approach allowed us to achieve a higher sequencing depth, which we considered important in the context of paired tumor samples. We agree that more comprehensive approaches, such as WES or WGS, could potentially reveal additional alterations. However, since some of the samples were FFPE tissue, this was not feasible. In a future study with a larger cohort, it would indeed be highly interesting to apply broader sequencing strategies to explore whether other genomic events may contribute to platinum resistance. We have now added this clarification to the Discussion (lines 742-749).
Provide additional details on sample selection and quality control, especially regarding excluded cases, to better understand potential biases.
We thank the reviewer for this important comment. All patients who underwent cytoreductive surgery at both diagnosis and relapse and had matched tumor tissue available were considered for inclusion. Of these, 7 cases were excluded due to insufficient tissue, low tumor cell content, or inadequate DNA quality for sequencing. Standard quality control procedures were applied, including histopathological assessment of tumor content, DNA quantification and integrity checks, as well as minimum sequencing depth requirements. We recognize that these selection criteria may introduce bias toward patients with higher-quality samples and longer survival, and we have now clarified these points in the Methods sections (line 102-105).
Statistical analysis is described but limited—consider clarifying whether survival comparisons (BRCA-mutated vs. wild-type) were formally tested.
We thank the reviewer for this helpful comment. In our Results, we report descriptive differences in time to relapse and overall follow-up between BRCA1/2-mutated and BRCA1/2 wild-type patients. Given the very limited cohort size, we did not perform formal statistical testing, as the study was not powered to detect significant subgroup differences. We have now clarified this in the Methods (lines 153-157) and emphasized that the reported numbers should be interpreted as descriptive and hypothesis-generating.
Results:
Results are clearly presented, but the small sample size (n=16) should be emphasized earlier as a key limitation.
We thank the reviewer for this important comment. We agree that the small sample size is a key limitation of our study. We have now emphasized this point earlier in the manuscript by adding a statement in the Introduction (lines 98-99), in addition to discussing it as a limitation later in the Discussion (lines 706-713).
The clinical outcomes section would benefit from more quantitative survival data (median OS, if estimable, or landmark survival rates).
We thank the reviewer for this valuable comment. Overall survival could not be reliably estimated in our cohort, as only 2 of 16 patients had died at the time of data cut-off, while the majority were still alive. For this reason, we did not report median OS. Instead, we chose to present time to first relapse and total follow-up time as clinically meaningful outcome measures. We have now clarified this in the Methods (lines 115-117).
The observation that BRCA-mutated patients had less carcinomatosis at relapse is interesting but underpowered; it should be framed strictly as hypothesis-generating.
We fully agree that the observation of less carcinomatosis at relapse among BRCA-mutated patients is based on very limited numbers and should be interpreted with caution. We have therefore revised the text to clearly frame this finding as descriptive and hypothesis-generating only (lines 289-293).
Discussion:
The discussion is thorough but could be more concise, as some sections repeat background information.
We thank the reviewer for this constructive comment. We have revised the Discussion to reduce redundancy and improve conciseness, while retaining the key points. Specifically, we have shortened sections that repeated background information describing BriTROC-1, OCTIPS and DESKTOP III.
Expand on the clinical implications: Should mutational profiling at relapse be deprioritized, and should resources shift toward studying epigenetic or non-genetic resistance mechanisms?
Thank you for this valuable comment. We agree that our findings support a critical reflection on the clinical utility of repeated mutational profiling at relapse. Given the high concordance between diagnostic and relapse samples, our data suggest that resources might be better directed toward investigating alternative resistance mechanisms, such as epigenetic changes, transcriptomic reprogramming, or alterations in the tumor microenvironment, which could underlie treatment resistance despite stable mutational profiles. We have expanded on these clinical implications in the Discussion section (lines 283-288).
The potential role of BRCA status in surgical selection is intriguing but requires cautious wording; suggest adding more context on how this could inform future predictive models.
Thank you for this insightful comment. We agree that the potential association between BRCA status and surgical outcomes at relapse should be interpreted with caution. We have now added contextualization to emphasize that our observation is hypothesis-generating and to outline how future research could explore whether BRCA status, as a surrogate of tumor biology, may contribute to predictive models for resectability in recurrent HGSC. This has been clarified in the Discussion (lines 307-311).
Limitations: the following should be stressed more:
Small, highly selected cohort—results may not generalize to unresectable patients.
Thank you for this important comment. We agree that the small and highly selected cohort represents a key limitation, and that our findings may not be generalizable to unresectable patients. We have now emphasized this point as the first limitation in the Discussion (lines 706-718) and expanded the description to clarify that the limited cohort size affects not only subgroup analyses but the overall interpretability of the findings.
Use of targeted sequencing rather than broader genomic or epigenomic profiling.
We agree that the use of targeted sequencing rather than broader genomic or epigenomic profiling represents a limitation of our study. We have now included this point in the Discussion (lines 742-748)
Potential bias from excluding cases with insufficient tumor tissue.
We agree that excluding cases with insufficient tumor tissue could introduce a potential selection bias. We have now emphasized this limitation in the Discussion (lines 735-737).
Minor Comments
Abstract: Could highlight the clinical takeaway more clearly (e.g., stability of mutations suggests limited utility of repeat sequencing at relapse).
We agree that the clinical takeaway should be more clearly emphasized. We have revised the Abstract accordingly to highlight the stability of the mutational profile and its clinical implication—specifically, that repeat mutational sequencing at relapse may have limited utility, and that future efforts should focus on non-genetic resistance mechanisms.
This is a well-conducted and carefully reported study that adds meaningful data on mutational stability in HGSC and long-term outcomes after cytoreductive surgery at relapse. With revisions to clarify novelty, strengthen methodological transparency, and streamline the discussion, the manuscript will make a valuable contribution.
We sincerely thank the reviewer for the positive and constructive feedback. We appreciate the recognition of our study’s contribution and have carefully addressed all comments to improve clarity, transparency, and focus. We believe the revised manuscript has been substantially strengthened and are grateful for the thoughtful review process.
Reviewer 2 Report
Comments and Suggestions for Authors
This manuscript investigates the molecular stability of high-grade serous ovarian cancer (HGSC) from diagnosis to relapse in a cohort of 16 patients who underwent cytoreductive surgery at both timepoints. Using a 501-gene sequencing panel, the study found remarkable stability in the mutational landscape, with only one patient acquiring a new mutation at relapse. Clinical outcomes, including high complete resection rates and favorable survival, are also described.
The study is clinically relevant and adds to the growing evidence that the mutational drivers of HGSC are established early and remain stable throughout disease progression. However, the study has notable limitations, particularly its small and highly selected cohort, which restricts generalizability. Below, I provide comments to help strengthen your manuscript:
Strengths
- The study is novel, providing paired sequencing data from both diagnosis and relapse.
- Clinical relevance is clear: your results support the rationale for secondary cytoreductive surgery and highlight the stability of genomic drivers in HGSC.
- Methodological rigor is commendable, with careful pathological review and standardized sequencing.
- Long-term follow-up (median 63 months) adds robustness to the clinical outcomes.
Limitation
- Sample Size: Only 16 patients included, limiting statistical power.
- Highly Selected Cohort: All underwent relapse surgery, biasing toward patients with better prognosis.
- Single-Center Design: Limits international applicability.
- Exclusion Bias: Interval debulking patients were more likely excluded due to tissue insufficiency.
- Outcome Measures: No formal survival analysis (e.g., OS, PFS) was performed despite long follow-up.
Major Points
- The study includes only 16 patients, all of whom underwent relapse surgery. Patients undergoing relapse cytoreductive surgery represent a favorable subgroup. The authors should caution against extrapolating findings to unresectable or platinum-resistant populations. The discussion should clearly emphasize this limitation and frame results as hypothesis-generating rather than conclusive.
- Since the cohort was followed for a median of 63 months, formal survival analyses (Kaplan–Meier curves for OS and PFS) would significantly strengthen the clinical relevance of your results.
- Please expand on how your findings align with or differ from BriTROC-1 and OCTIPS studies. More explicit comparisons would highlight the value of your dataset.
- The manuscript emphasizes genomic stability but does not sufficiently expand on alternative mechanisms (epigenetic, microenvironmental, transcriptomic). This could be elaborated in the discussion.
- The study design reflects national practices and patient selection in Denmark. Please address how these findings may or may not apply internationally.
Minor Points
- Consider adding graphical or tabular visualizations that show mutational concordance between diagnosis and relapse.
- “Partial platinum-sensitive” should be consistently defined and used.
- Supplementary tables (S1–S4) are valuable but should be more clearly referenced in the Results section.
- Please clarify whether BRCA mutations reported were somatic only, or whether germline testing was available.
This is a well-conducted study that addresses an important question in recurrent HGSC. The findings provide evidence of genomic stability across disease course and highlight the potential of secondary cytoreductive surgery in carefully selected patients. However, it requires expanded discussion of its limitations, inclusion of formal survival analysis, and clearer contextualization with external datasets before being suitable for publication.
Author Response
Major Points
- The study includes only 16 patients, all of whom underwent relapse surgery. Patients undergoing relapse cytoreductive surgery represent a favorable subgroup. The authors should caution against extrapolating findings to unresectable or platinum-resistant populations. The discussion should clearly emphasize this limitation and frame results as hypothesis-generating rather than conclusive.
Thank you for this important comment. We agree that our cohort represents a highly selected and surgically favorable subgroup, and that the findings should not be extrapolated to unresectable or platinum-resistant patients. We have now emphasized this limitation more clearly in the Discussion (lines 706-718), highlighting that the results should be interpreted with caution and regarded as hypothesis-generating observations to be validated in broader, less selected cohorts.
- Since the cohort was followed for a median of 63 months, formal survival analyses (Kaplan–Meier curves for OS and PFS) would significantly strengthen the clinical relevance of your results.
Thank you for this valuable suggestion. We agree that formal survival analyses could provide additional clinical context. However, at the time of data cut-off, only two patients had died, which precluded meaningful Kaplan–Meier analysis for overall survival. Similarly, progression-free survival was not calculated, as the number and timing of relapses varied substantially between patients. Instead, we reported time to first relapse and total follow-up time as clinically meaningful and comparable endpoints for this specific cohort. We have clarified this rationale in the Methods (lines 115-117).
- Please expand on how your findings align with or differ from BriTROC-1 and OCTIPS studies. More explicit comparisons would highlight the value of your dataset.
Thank you for this helpful comment. We agree that a clearer comparison with the BriTROC-1 and OCTIPS studies strengthens the context and highlights the value of our dataset. We have now expanded this section of the Discussion (lines 234-264) to include more explicit comparisons, outlining similarities and key differences in study design, patient selection, and findings.
- The manuscript emphasizes genomic stability but does not sufficiently expand on alternative mechanisms (epigenetic, microenvironmental, transcriptomic). This could be elaborated in the discussion.
Thank you for this helpful comment. We agree that non-genetic mechanisms likely play an important role in treatment resistance. We have expanded the Discussion (lines 284-288) to highlight the potential relevance of epigenetic, transcriptomic, and microenvironmental alterations as areas for future research.
- The study design reflects national practices and patient selection in Denmark. Please address how these findings may or may not apply internationally.
Thank you for this important comment. We agree that national practice patterns may limit direct generalization to other settings. We have now elaborated in the Discussion (lines 718-724) to note that treatment selection and surgical criteria may vary internationally, while emphasizing that Denmark follows ESMO–ESGO guidelines, making our findings broadly comparable to other Western healthcare systems.
Minor Points
- Consider adding graphical or tabular visualizations that show mutational concordance between diagnosis and relapse.
Thank you for this helpful suggestion. We agree that a graphical representation can be useful for illustrating mutational concordance. However, because our cohort is small and we aimed to report the exact gene variants observed at diagnosis and relapse (e.g., BRCA2 c.7876T>C p.Trp2626Arg exon 17), we considered a tabular format to be more informative and transparent. Table 2 therefore provides a detailed overview of mutational concordance between diagnosis and relapse, including specific gene-level alterations, which would not be fully captured in a graphical summary.
- “Partial platinum-sensitive” should be consistently defined and used.
We agree that consistent terminology is important. Partial platinum sensitivity was defined in the Methods section as relapse occurring between 6 and 12 months after treatment completion, and this definition has been applied consistently throughout the manuscript (see Methods, lines 112-114, and Results, lines 168-169).
- Supplementary tables (S1–S4) are valuable but should be more clearly referenced in the Results section.
Thank you for this helpful comment. We agree that clear referencing of the supplementary tables is important for readability. The supplementary material (Tables S1–S4) is referenced at relevant points throughout the Results section, each in direct relation to the corresponding findings We have double-checked the manuscript to ensure that these references are clear and consistent.
- Please clarify whether BRCA mutations reported were somatic only, or whether germline testing was available.
Thank you for this important comment. All molecular analyses performed in this study were somatic and not germline. We have now clarified this in the Methods section (lines 136-137).
This is a well-conducted study that addresses an important question in recurrent HGSC. The findings provide evidence of genomic stability across disease course and highlight the potential of secondary cytoreductive surgery in carefully selected patients. However, it requires expanded discussion of its limitations, inclusion of formal survival analysis, and clearer contextualization with external datasets before being suitable for publication.
We sincerely thank the reviewer for the positive and constructive feedback. We appreciate the acknowledgment of our study’s design and relevance. In response to the reviewer’s comments, we have expanded the Discussion to more clearly address study limitations and clarified the generalizability of our findings. Formal survival analyses were not feasible due to the limited number of events, but this rationale is now explicitly stated in the Methods and Results sections. We believe these revisions have strengthened the manuscript and improved its clarity and interpretability.
Round 2
Reviewer 1 Report
Comments and Suggestions for Authors
The authors have addressed all my concerns adequately and I believe the manuscript can be published in the current form.
Reviewer 2 Report
Comments and Suggestions for Authors
Thank you for your thorough and thoughtful revision. The manuscript has been significantly improved, and all major and minor comments from the previous review have been carefully addressed. The rationale, methodology, and discussion are now clear and well balanced.
The integration of comparisons with prior studies such as BriTROC-1, OCTIPS, and DESKTOP III provides valuable context, and the expanded discussion on mutational stability and alternative mechanisms (epigenetic or microenvironmental) adds important scientific depth. The limitations are appropriately acknowledged, and the inclusion of detailed supplementary tables enhances transparency.
Overall, the revised version is well written, coherent, and scientifically sound. I consider this paper worthy of publication in this journal after review by the editorial office.